# Collaborative mapping: Perceived environmental risk of leptospirosis in Urban communities in Salvador, Brazil

Fabiana Almerinda G. Palma[1][⍟], Pablo Ruiz Cuenca[2][⍟], Hussein Khalil[3], Patrícia Lustosa Brito[4], Marbrisa Nascimento Reis das Virgens[4], Murilo Guerreiro Arouca[5], Alexandre Mota Santos[1], Daiana Santos de Oliveira[1,6], Yeimi Alexandra Alzete Lòpez[1], Diogo César de C. Santiago[7], Michael Begon[8], Emanuele Giorgi[2], Federico Costa[1,6]*, Ricardo Lustosa[9][⍟]

1 Institute of Collective Health, Federal University of Bahia, Canela, Salvador, Bahia, Brazil, 2 Centre for Health Informatics, Computing and Statistics (CHICAS), Health Innovation One, Lancaster University, Lancaster, United Kingdom, 3 Department of Wildlife, Fish, and Environmental Studies, Swedish University of Agricultural Sciences, Umeå, Sweden, 4 Polytechnic School, Federal University of Bahia, Salvador, Bahia, Brazil, 5 Institute of Computing, Federal University of Bahia, Salvador, Bahia, Brazil, 6 Instituto Gonçalo Moniz, Fundação Oswaldo Cruz, Ministério da Saúde, Salvador, Bahia, Brazil, 7 Departamento de Medicina Preventiva e Social, Faculdade de Medicina da Bahia, Federal University of Bahia, Salvador, Brazil, 8 Department of Evolution, Ecology and Behaviour, The University of Liverpool, Liverpool, United Kingdom, 9 Federal University of Western Bahia - Campus Barra, Bahia, Brazil

⍟ These authors contributed equally to this work.
* e.giorgi@lancaster.ac.uk

## Abstract

Environmental risks disproportionately affect socially vulnerable populations, yet most studies on urban leptospirosis have not captured residents' perceptions of these risks. We conducted a cross-sectional study between August 2018 and March 2019 across four communities in Salvador, Brazil (Marechal Rondon, Alto do Cabrito, Nova Constituinte, and Rio Sena) using a collaborative mapping approach to explore residents' perceptions of environmental health risks. The majority of participants identified sewage as the primary health risk (26–39% across communities), followed by garbage (22–28%), while leptospirosis itself was perceived as a high-risk problem by very few participants (0–4%). We found no spatial concordance between areas perceived as high risk for leptospirosis and households testing positive for the disease, nor between areas perceived as high rat risk and the locations of positive rat traps and track plates. However, a higher density of rubbish piles was observed in areas residents identified as high rubbish risk, suggesting that perceptions do reflect some on-the-ground realities. Collaborative mapping proved an effective method for capturing individual perceptions of environmental health risks and promoting community participation in generating locally relevant information. The disconnect between perceived and actual transmission risk may reflect that residents are more attuned to the broader social and environmental determinants of leptospirosis, such as sanitation and waste, than to the disease itself, a distinction that warrants further

**Data availability statement:** The datasets used and/or analyzed during the current study cannot be shared publicly because of personal information of participants in the survey. Researchers who wish to access the data can contact the Research and Extension Support Center – Multidisciplinary Center of Barra / Federal University of Western Bahia (UFOB) using nape.barra@ufob.edu.br.

**Funding:** FC was funded by Medical Research Council (UK) (Grant number: MR/ P024084/1). The funder had no role in the design, data collection, analysis, decision to publish, or preparation of manuscript. PRC received a PhD studentship funded by the Medical Research Council UK.

**Competing interests:** The authors have declared that no competing interests exist.

investigation to better integrate community knowledge into disease prevention and control strategies.

## Introduction

Environmental risks pose a significant threat to populations in situations of social vulnerability, exposing them to a wide range of zoonotic diseases [1]. Leptospirosis, a neglected zoonosis [2] transmitted through contact with contaminated water and mud [3], is a significant health risk and is strongly influenced by environmental factors [1,4,5], such as waste and poor sewage systems, both of which result from inadequate basic sanitation [6]. Previous studies using quantitative and spatial methodologies have reported various environmental risk factors for leptospirosis, such as residences in flood-prone areas, proximity to open sewage, sightings of rats, contact with mud, and inadequate sanitation [5,7]. Despite their importance, these studies have not considered the perceptions of people living in areas with these risks and the influence of their personal experiences, those of other family members, and the community on the characteristics of their territory, as well as media reports [8].

In Brazil, there is a need to understand the perception of risk by residents living in communities exposed to a wide range of preventable environmental risk factors that cause diseases such as leptospirosis. Using participatory methods to determine residents' perceptions based on a critical view of the environment can contribute to people living in these contexts' awareness and engagement [9]. This is especially needed in populations exposed to numerous environmental risks, which tend to be amplified by population growth in areas with inadequate infrastructure and sanitation [6], and the emergence and re-emergence of diseases associated with environmental risk factors [10].

Assessing environmental risk perception to mitigate its effects on human life and reduce the risk of diseases can be accomplished by adopting participatory methodologies such as collaborative mapping. This method promotes community participatory work and the development of an understanding of existing local resources and the environmental problems that affect them [11]. Collaborative mapping activities are guided by the environmental context and how groups interpret their environment based on their daily experiences [12]. This allows residents and researchers to reflect more deeply on the environment being studied [13] and identify more effective and sustainable ways to intervene in existing risks. Previous studies using collaborative mapping have contributed to identifying environmental risks such as floods [14], volcanic risk [15], flooding and landslides [16], mangrove ecosystem services [17], and measured the risk environment of female sex workers in the Dominican Republic [18]. The participatory approach can be fundamental for creating responses, indicating community-based interventions, and providing the community with access to information that can support claims for public policies aimed at necessary improvements in their environment.

In most studies conducted in urban informal settlements, traditional spatial methodologies have been used to identify environmental risk factors for leptospirosis

[5,7]]. The methods have involved recall from residents, but have not captured their perceptions of the surrounding environment. Capturing these perceptions could be a useful tool in understanding how the community understands the environmental risks they are exposed to and could enhance researchers understanding of the links between environmental factors and leptospirosis risk. Therefore, we aim to analyze the perception of environmental health risks through collaborative mapping, focusing on the factors contributing to leptospirosis in four peripheral communities of Salvador, Brazil.

## Materials and methods

### Ethics statement

This study was approved by the Research Ethics Committee of the Collective Health Institute/ Federal University of Bahia (CEP/ISC/UFBA), with CAAE number 68887417.9.0000.5030, and by the National Research Ethics Committee (CONEP) linked to the Brazilian Ministry of the Health under approval numbers 2.245.914–2.245.914.17–3.315.568. Informed consent was obtained from all participants and/or their legal guardian/s.

### Study area

Our study was part of a project initiated in 2017 in four urban informal settlements [1]]. These communities, known as Marechal Rondon (MR), Alto do Cabrito (AC), Nova Constituinte (NC), and Rio Sena (RS), are situated in the periphery of Salvador, a city of 2.417.678 million inhabitants in 2022 [19]] in Bahia, Northeast Brazil. (Fig 1). They have similar characteristics: low family income, densely populated settlements; and precarious infrastructure, sanitation, and public service offerings, such as healthcare and public transport [6]]. Furthermore, previous studies in this area indicated that an unadjusted seroprevalence for *Leptospira* in MR, AC, NC and RS was 11% (n = 338), 10% (n = 375), 9% (n = 306), 12% (n = 299), respectively [1]], which characterizes these sites as being at risk for leptospirosis transmission.

### Serological data

A serological analysis was performed on consenting participants across all four study areas using the Microscopic Agglutination Test (MAT), as described in Khalil [1]] and De Oliveira [20]]. MAT is the gold-standard diagnostic method for leptospirosis and is based on dark-field microscopy detection of serum agglutination samples from an individual with live *Leptospira* antibodies.

Samples were tested against a panel of seven Leptospira strains, including five international reference strains (WHO Collaborative Laboratory for Leptospirosis, Royal Tropical Institute, Holland): L. kirschneri serogroups Cynopteri (strain 3522C) and Grippotyphosa (strain Duyster); L. interrogans serogroups Canicola (strain H. Utrecht IV) and Autumnalis (strain Akiyami A); and L. borgpetersenii serogroup Ballum (strain MUS 127). The panel also included two local clinical isolates: L. interrogans serogroup Icterohaemorrhagiae (serovar Copenhageni, strain Fiocruz L1-130) and L. santarosai serogroup Shermani (strain LV3954).

These serogroups were selected because they represent both widely used reference antigens and locally prevalent isolates, as reported in previous investigations [1,20]. Samples reactive at a 1:100 dilution were further titrated in serial twofold dilutions. Participants were considered seropositive if reactive to one or more serogroups, including mixed reactions. Titers <1:50 were considered negative.

### Collaborative mapping

The collaborative mapping process was carried out in three phases, detailed below.

**Phase 1: Recruitment of young partners.** Two cultural associations from the study areas, the Emília Machado Bahia group (AEMBA) in Marechal Rondon and Centro Cultural Teatro E² in Alto do Cabrito, were approached to recruit young residents to be involved as research partners. These young partners were trained to take part in fieldwork and mapping

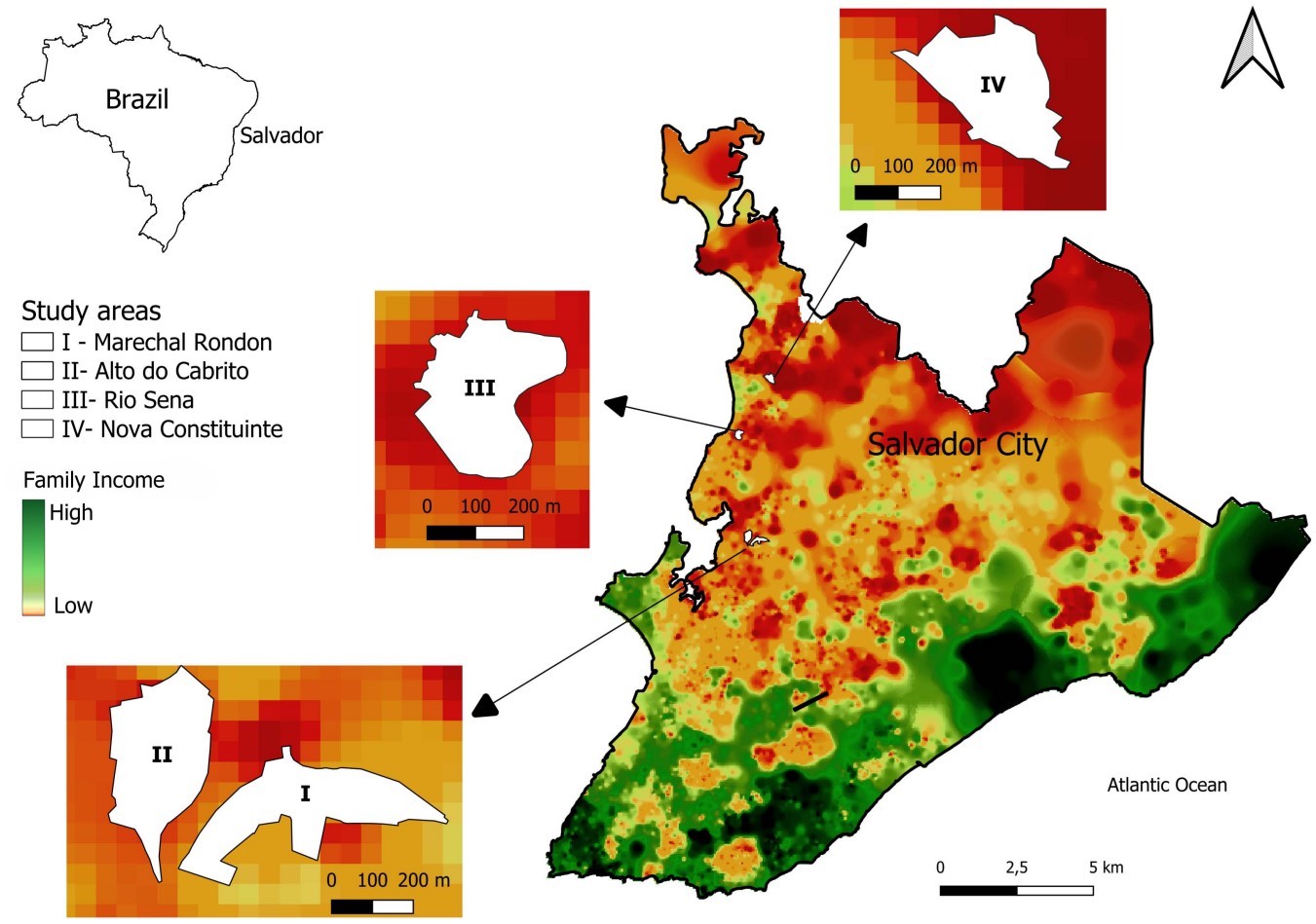

**Fig 1. Map showing the location of the four study areas: Marechal Rondon (I), Alto do Cabrito (II), Rio Sena (III) and Nova Constituinte (IV), outlined in white in Salvador, Brazil.** Distribution of family income, high (green) and low (red). The base map used in this figure was obtained from the territorial boundary shapefiles provided by the Brazilian Institute of Geography and Statistics (IBGE), available at: https://www.ibge.gov.br/geociencias/organizacao-do-territorio/malhas-territoriais.html. The IBGE geospatial datasets are publicly available as open government data. Information about data access and reuse is available at: https://www.ibge.gov.br/acesso-informacao/dados-abertos.html.

activities. The criteria for recruitment included being aged between 13 and 22 and participating in the activities of local organizations. The final group of young partners had to be composed of at least 50% self declared black or mixed race individuals.

**Phase 2: Field surveys.** Teams were created to carry out field surveys. These were made up of one young local research partner, one agent from the Zoonosis Control Center (CCZ) and one university researcher with experience in spatial data collection. Targeted sampling from the serological survey of 244 residents across all four study areas was performed, using cataloged households from a 2018 census [1]]. Residents able to participate were individuals who were aged >18 years old, slept three nights per week within the study area, were able to provide written informed consent and were considered heads of household were asked to complete a questionnaire. These questionnaires captured the participant's environmental health risk perception and were carried out between August 2018 and March 2019.

The questionnaire asked the participant to name the main environmental risks to health that were present in the community. The participant was then asked to draw on a map of where the risks were. We used a colored A3 map with

satellite images from the Google Earth platform at a scale of 1/500 meters. The participant was first introduced to local landmarks on the map, such as their own homes and streets, to allow them to locate themselves on the image. Then, they were asked to draw the specific risks they had mentioned on the map and give them a level of risk (low, medium or high). They could draw these risks as points, lines or polygons, creating georeferrenced risk perceptions. Participants could assign multiple risks to the same area they had identified on the map.

These georeferrenced risk perceptions were processed using the Vicon SAGA Geographic Information System platform [21]]. The young research partners digitized the perceptions drawn on the maps, creating shapefiles of each answer given by each participant.

**Phase 3: Spatial analysis.** The georeferrenced perception data was processed to create a combined set of perceptions for each study area. Perceptions that were stored as lines or points were given a 5 meter circular buffer around them, due to spatial precision uncertainty, before being merged with polygons. These mapped perceptions were then grouped by type and level of risk perceived. Given the sparse use of all three risk categories (Table 3), low and medium perceived risk were combined into one group. This created a unique area for each type of risk perception, level of risk (high and low) and study area. We developed these categories of risk through discussions and consensus with experts in the fields of ecology, epidemiology, veterinary medicine and young people from the communities.

The different perceptions were then compared to objective measures of risk. Households in each of the study areas were classified into two groups: those with at least one known leptospirosis sero-positive individual living there and those with none, using the serological data described above. This was used as a proxy for leptospirosis infection risk across the study areas. Rat traps and rat track plates were used as objective measures of rat risk. Traps that had caught at least one rat and track plates that were used by rats were both considered positive measures of rat presence. Matching pairs (e.g., high risk perception and leptospirosis positive households, low risk and negative households) were compared to incongruous pairs (e.g., high risk and negative households, low risk and positive households) to create a Cohen's kappa score for agreement. To facilitate the analysis, high risk perception took precedence over low risk perception in areas where both were present. The locations of rubbish piles were used to calculate the density of these with each type of perception polygon.

More details information on the data collection process can be found in S2 Appendix.

### Software and packages

The map used in the publication is not copyrighted, as it was created by the authors themselves using the free and open-source software R. The data used in preparing the map containing the boundaries of neighboring countries, Brazil, Bahia, Salvador, and neighborhoods, are also open data and were downloaded from the website of the Instituto Brasileiro de Geografia e Estatística (IBGE, 2021) and the Prefeitura Municipal de Salvador (Salvador, 2017). Both data can be consulted with their respective metadata on the websites: https://www.ibge.gov.br/geociencias/organizacao-do-territorio/malhas-territoriais/15774-malhas.html?=&t=downloads; http://mapeamento.salvador.ba.gov.br/geo/desktop/index.html#on=layer/default;bairros/bairros;scalebar_meters/scalebar_m;orto2016/Ortoimagem_Salvador_2016_2017&loc=76.43702828517625;-4278080;-1445884.

All statistical analyses were carried out in R version 4.5.1. We used packages from the {tidyverse} [22], {sf} [23] and {terra} [24] for data wrangling and {caret} [25] and {vcd} [26] to calculate the Kappa scores.

## Results

### Sociodemographic and serological characteristics

Among the eligible residents based on the study's inclusion criteria (see Phase 2: Field surveys), 244 participants from the four communities were included in the study. Of these, the majority were female (70.5%), with a relatively uniform

distribution between the communities: AC (71.6%), MR (64.9%), NC (72.7%), and RS (72.5%). The mean age of the participants was 46.6 years (standard deviation, SD = 15.2), varying between 44.0 years in AC (SD = 14.8) and 51.9 years in MR (SD = 16.6). The majority of participants self-declared as black (48.4%) or mixed race (39.8%), a similar composition in each of the communities (Table 1).

Regarding occupation, the majority of participants were unemployed (67.6%), with the highest proportion of unemployed in RS (72.5%) and the lowest in AC (59.3%). The overall employment rate was 27.5%, highest in AC (29.6%) and lowest in MR community (22.8%). The average monthly income was R$537.6 (SD = 627.7), ranging from R$406.3 (SD = 535.7) in RS to R$584.9 (SD = 638.9) in AC (Table 1).

Most participants in this study consented to providing blood samples (91.8%) in the serological survey. In NC, all participants provided samples (100%), followed by RS (98.0%), AC (85.2%), and MR (87.7%). Refusal to provide blood samples was 3.3% of participants (Table 1).

The prevalence of leptospirosis antibodies across all study areas was 12.2%. This was highest in RS, where 19.6% of participants were serologically positive for *Leptospira* antibodies, and lowest in AC with 8.6%. However, there were a

**Table 1. Sociodemographic characteristics of participants in study areas.**

| Sociodemographic characteristics | | Communities | | | |
|---|---|---|---|---|---|
| | Overall | AC | MR | NC | RS |
| | N = 244 | N = 81 | N = 57 | N = 55 | N = 51 |
| | Frequency (%) or Mean (SD) | | | | |
| **Sex** | | | | | |
| Female | 172 (70.5) | 58 (71.6) | 37 (64.9) | 40 (72.7) | 37 (72.5) |
| Male | 72 (29.5) | 23 (28.4) | 20 (35.1) | 15 (27.3) | 14 (27.5) |
| **Age (years)** | 46.6 (15.2) | 44.0 (14.8) | 51.9 (16.6) | 47.2 (14.9) | 44.1 (13.0) |
| *Missing* | 1 (0.4) | 1 (1.1) | 0 (0.0) | 0 (0.0) | 0 (0.0) |
| **Ethnicity** | | | | | |
| White | 10 (4.1) | 3 (3.7) | 2 (3.5) | 2 (3.6) | 3 (5.9) |
| Black | 118 (48.4) | 38 (46.9) | 31 (54.4) | 27 (49.1) | 22 (43.1) |
| Asian | 5 (2.0) | 2 (2.5) | 2 (3.5) | 1 (1.8) | 0 (0.0) |
| Mixed | 97 (39.8) | 27 (33.3) | 19 (33.3) | 25 (45.5) | 26 (51) |
| Indigenous | 2 (0.8) | 2 (2.5) | 0 (0.0) | 0 (0.0) | 0 (0.0) |
| *Missing* | 12 (4.9) | 9 (11.1) | 3 (5.3) | 0 (0.0) | 0 (0.0) |
| **Current employment** | | | | | |
| Unemployed | 165 (67.6) | 48 (59.3) | 41 (71.9) | 39 (70.9) | 37 (72.5) |
| Employed | 67 (27.5) | 24 (29.6) | 13 (22.8) | 16 (29.1) | 14 (27.5) |
| *Missing* | 12 (4.9) | 9 (11.1) | 3 (5.3) | 0 (0.0) | 0 (0.0) |
| **Monthly salary (R$)** | 537.6 (627.7) | 584.9 (638.9) | 566.9 (527.6) | 564.1 (765.5) | 406.3 (535.7) |
| *Missing* | 19 (7.8) | 11 (13.6) | 4 (7.0) | 1 (1.8) | 3 (5.8) |
| **Blood sample** | | | | | |
| Yes | 224 (91.8) | 69 (85.2) | 50 (87.7) | 55 (100) | 50 (98.0) |
| No | 8 (3.3) | 3 (3.7) | 4 (7.0) | 0 (0.0) | 1 (2.0) |
| *Missing* | 12 (4.9) | 9 (11.1) | 3 (5.3) | 0 (0.0) | 0 (0.0) |
| **Leptospirosis serological status** | | | | | |
| Positive | 30 (12.2) | 7 (8.6) | 6 (10.5) | 7 (12.7) | 10 (19.6) |
| Negative | 194 (79.5) | 62 (76.5) | 44 (77.2) | 48 (87.3) | 40 (78.4) |
| *Missing* | 20 (8.2) | 12 (14.8) | 7 (12.3) | 0 (0.0) | 1 (2.0) |

number of individuals who did not consent to have their blood sample taken and for which we could not ascertain their serological status.

Errors during the data collection phase resulted in some socio-demographic characteristics not being stored appropriately. This resulted in missing data, as is reported in Table 1.

## Environmental characteristics

Households which provided perception data were compared to all households that were censured in the respective study areas (Table 2). We have also included a column showing the environmental characteristics of leptospirosis positive households to add further context to the study areas. These are defined as having at least one serologically positive individual living in the house. Households sampled in the present study in AC, MR, and RS were surrounded by more vegetation (910.5 m² [AC], 600.2 m² [MR], and 1473.0 m² [RS]) and soil (79.6 m² [AC], 83.1 m² [MR], and 38.3 m² [RS]) and less impervious land (1835.6 m² [AC], 2130.1 m² [MR], and 1314.2 m² [RS]) compared to the community mean (vegetation: 801.7 m² [AC], 525.6 m² [MR], and 1212.4 m² [RS]; soil: 71.5 m² [AC], 86.1 m² [MR], and 35.8 m² [RS]; impervious: 1952.9 m² [AC], 2201.8 m² [MR], and 1577.4 m² [RS]). The opposite was seen in NC, where sample households were located in areas with more impervious land (2055.3 m² vs. 1885.9 m²) than the community mean (Table 2).

Similarly, in these three study areas, the sample households were found at closer distances to sewers (172.4 m [AC], 83.0 m [MR], and 73.4 m [RS]) and lower elevations (57.2 m [AC], 48.5 m [MR], and 63.1 m [RS]) than the community mean (distance to sewer: 173.3 m [AC], 93.7 m [MR], and 74.5 m [RS]; elevation: 58.9 m [AC], 51.2 m [MR], and 66.7 m [RS]). Sample households in AC and MR were found at farther distances to rubbish piles (46.0 m [AC] and 28.9 m [MR])

Table 2. Environmental characteristics of households in study areas.

| Environmental characteristics | | Communities | | | | | | | | | | | |
| --- | --- | --- | --- | --- | --- | --- | --- | --- | --- | --- | --- | --- | --- |
| | | AC | | | MR | | | NC | | | RS | | |
| | | All house-holds | Sample house-holds | Leptospiro-sis positive households | All house-holds | Sample house-holds | Leptospiro-sis positive households | All house-holds | Sample house-holds | Leptospiro-sis positive households | All house-holds | Sample house-holds | Leptospiro-sis positive households |
| | | N=610 | N=103 | N=34 | N=977 | N=121 | N=33 | N=484 | N=69 | N=24 | N=439 | N=67 | N=33 |
| | | Mean (SD) | Mean (SD) | Mean (SD) | Mean (SD) | Mean (SD) | Mean (SD) | Mean (SD) | Mean (SD) | Mean (SD) | Mean (SD) | Mean (SD) | Mean (SD) |
| Land cover (m²)[1] | Impervious | 1952.9 (396.3) | 1835.6 (421.2) | 1771.9 (354.1) | 2201.8 (415.1) | 2130.1 (470.2) | 2127.2 (413.5) | 1885.9 (510.7) | 2055.3 (413.2) | 1825.5 (462.9) | 1577.4 (611.4) | 1314.2 (554.0) | 1464.6 (639.2) |
| | Vege-tation | 801.7 (391.1) | 910.5 (428.9) | 980.2 (359.6) | 525.6 (370.6) | 600.2 (450.9) | 594.6 (363.7) | 832.9 (446.2) | 706.7 (369.7) | 929.7 (352.3) | 1212.4 (607.9) | 1473.0 (552.9) | 1332.2 (635.8) |
| | Soil | 71.5 (88.4) | 79.6 (90.8) | 72.0 (81.8) | 86.1 (107.0) | 83.1 (92.3) | 90.7 (107.6) | 106.0 (199.5) | 62.6 (157.0) | 66.8 (166.6) | 35.8 (53.5) | 38.3 (59.1) | 28.6 (51.0) |
| | Water | 1.3 (9.0) | 1.7 (10.9) | 3.3 (16.5) | 12.9 (34.3) | 12.7 (30.2) | 14.9 (29.3) | 2.7 (16.4) | 2.8 (14.3) | 5.4 (21.7) | 1.8 (5.2) | 2.0 (5.3) | 2.0 (4.6) |
| Elevation (m above sea level) | | 58.9 (10.6) | 57.2 (9.9) | 53.7 (9.2) | 51.2 (9.0) | 48.5 (7.8) | 48.0 (7.3) | 7.5 (4.6) | 7.9 (4.6) | 10.6 (5.9) | 66.7 (15.2) | 63.1 (12.7) | 66.1 (12.1) |
| Distance to sewer (m) | | 173.3 (71.4) | 172.4 (65.7) | 192.9 (75.8) | 93.7 (69.5) | 83.0 (66.9) | 71.8 (62.9) | 123.6 (65.0) | 135.5 (69.9) | 152.9 (71.1) | 74.5 (44.3) | 73.4 (44.2) | 62.2 (43.2) |
| Distance to rubbish piles (m) | | 41.6 (27.3) | 46.0 (27.3) | 48.5 (29.3) | 26.7 (19.5) | 28.9 (21.9) | 34.5 (17.2) | 34.6 (24.7) | 30.2 (22.2) | 35.3 (25.8) | 43.0 (28.8) | 42.2 (26.2) | 41.2 (26.4) |

[1]Land cover classifications are mutually exclusive and show the area, in m², of each kind of cover within a 30m circular buffer around each household.

compared to the community mean (41.6 m [AC] and 26.7 m [MR]), whilst those in NC and RS were closer (30.2 m [NC] and 42.2 m [RS]) compared to the community mean (34.6m [NC] and 43.0 m [RS]) (Table 2).

In general, leptospirosis positive households had less impervious land surrounding them but more vegetation and soil than the community average. They were also located at lower elevations, except for in NC. This fits the current understanding of how peridomestic environment may be linked to an individuals' risk for leptospirosis.

## Perceived environmental health risk

Most participants perceived sewage, garbage, and rats as the primary environmental health risks. In all communities, participants identified sewage as the primary high-risk health issue (AC: 33.6%, MR: 32.5%, NC: 27.5%, and RS: 39.2%, percentages of all perception polygons for each area). Garbage was the second most frequently perceived problem, as a high health risk, by participants in the communities of AC (28.1%), RS (27.5%), MR (27.2%), and NC (21.5%) (Table 3).

Across all study areas, a lower proportion of participants perceived leptospirosis (1.4%) as a high risk health problem, which was also observed in the communities of MR (1.0%), NC (3.8%), and RS (2.5%). Furthermore, in AC, no participant perceived leptospirosis as a health risk problem (Table 3).

Additionally, there were more low risk perception polygons in NC compared to the other study areas. This was particularly true for sewage risk, where low risk for sewage made up 20% of all perception polygons for that area, compared to 1.3% in AC and RS, and 2.1% in MR. This effect could also be seen with perception of low risk for garbage (NC = 6.25%, AC = 2.1%, MR = 3.7%, RS = 3.8%) and low risk for rats (NC = 3.8%, AC = 0%, MR = 2.1%, RS = 1.3%) (Table 3).

## Perceived and objective environmental risk

There was no agreement across all perceived environmental risk factors and leptospirosis risk (Kappa scores: -0.02 [rubbish], -0.01 [sewage], 0.02 [rat] and -0.09 [leptospirosis]; Table 4 and Fig 2). The majority of households across all perception types fell within the leptospirosis negative and high risk perception pairing (rubbish: 154/245, sewage: 207/314,

Table 3. Environmental health risks, by perception polygons.

| Perceived risk* | Risk level | Overall | Communities | | | |
|---|---|---|---|---|---|---|
| | | | AC | MR | NC | RS |
| | | N = 496 | N = 146 | N = 191 | N = 80 | N = 79 |
| | | Frequency (%)† | | | | |
| Leptospirosis | Low | 4 (0.8) | 0 (0.0) | 1 (0.5) | 1 (1.3) | 2 (2.5) |
| | Medium | 1 (0.2) | 0 (0.0) | 1 (0.5) | 0 (0.0) | 0 (0.0) |
| | High | 7 (1.4) | 0 (0.0) | 2 (1.0) | 3 (3.8) | 2 (2.5) |
| Rat | Low | 8 (1.6) | 0 (0.0) | 4 (2.1) | 3 (3.8) | 1 (1.3) |
| | Medium | 25 (5.0) | 9 (6.2) | 3 (1.6) | 7 (8.8) | 6 (7.6) |
| | High | 79 (15.9) | 27 (18.5) | 29 (15.2) | 9 (11.3) | 14 (17.7) |
| Sewage | Low | 23 (4.6) | 2 (1.3) | 4 (2.1) | 16 (20) | 1 (1.3) |
| | Medium | 42 (8.5) | 11 (7.5) | 14 (7.3) | 5 (6.25) | 12 (15.2) |
| | High | 164 (33.1) | 49 (33.6) | 62 (32.5) | 22 (27.5) | 31 (39.2) |
| Garbage | Low | 18 (3.6) | 3 (2.1) | 7 (3.7) | 5 (6.25) | 3 (3.8) |
| | Medium | 46 (9.3) | 21 (14.4) | 16 (8.4) | 5 (6.25) | 4 (5.1) |
| | High | 125 (25.2) | 41 (28.1) | 52 (27.2) | 15 (18.75) | 17 (21.5) |

* Perceived risks are not mutually exclusive, some perception polygons had various risks assigned to them.

† Percentage of all perception polygons for each study area.

**Table 4. Number of households, characterised by their *Leptospira* seropositivity, within each type of risk perception polygon. Corresponding Kappa score showing agreement between perceptions and infected households.**

| Household leptospirosis status | Risk Perception | | | | | | | |
|---|---|---|---|---|---|---|---|---|
| | Garbage | | Sewage | | Rat | | Leptospirosis | |
| | Risk level | | | | | | | |
| | High | Low | High | Low | High | Low | High | Low |
| Positive | 42 | 13 | 65 | 11 | 50 | 9 | 4 | 5 |
| Negative | 154 | 36 | 207 | 31 | 162 | 36 | 27 | 16 |
| Kappa score | -0.02 | | -0.01 | | 0.02 | | -0.09 | |
| P value | 0.465 | | 0.752 | | 0.590 | | 0.331 | |

rat: 162/257, leptospirosis: 27/52). Similarly, the pairing with the lowest count was leptospirosis positive households and low risk perception for rubbish (13/245), sewage (11/314) and rat (9/257). Positive households and high risk perception for leptospirosis ranked the lowest for this perception group (4/25) (Table 4).

Comparing rat presence and rat risk perception, there was strong evidence of a slight disagreement, which was unlikely to have happened by chance (Table 5). The areas where people were perceiving as being high risk for rats had more negative traps and track plates across all study areas. Maps are available in Fig A in S1 Appendix.

Individuals were perceiving a high risk for rubbish where there were more rubbish piles. This created a higher density of rubbish piles in high risk perception polygons than in low risk perception polygons (Table 6). Maps are available in Fig B S1 Appendix.

## Discussion

In this study, we conducted extensive collaborative mapping in four urban informal settlements in Salvador, Brazil. Residents identified sewage, garbage, and rats as the main environmental health risks. However, there was no agreement between the areas perceived as high risk and the location of leptospirosis-positive households. Similarly, there was no agreement between the areas considered low-risk and leptospirosis-negative households.

Leptospirosis has been considered a disease that primarily affects adult men [5,7]], self-identified as Black (*preto* and *pardo*), living in regions located in valleys (low altitudes) with poor sewage services and infrastructure [5], [7]]. Vegetation and ground cover have been associated with an increased risk of contamination by *Leptospira*, and consequently, an increased risk of human infection [7]]. The predominant profile of participants in our study was that of unemployed women with low income. Consequently, the risk perception results primarily reflect the risk perceptions of females, which may differ from those of males. One issue that stands out in this study is that those participants did not identify flooding as a risk in their communities, despite it being a risk factor for leptospirosis [7]. We also found that households included in our study were surrounded by more vegetation and soil, and less impervious terrain, in addition to being closer to sewage systems and located at lower altitudes when compared to all households in the respective communities.

The population profile found in this study, despite some missing data, may have implications for the perception of environmental risks. For example, adults generally tend to have a greater identification with their place of residence compared to younger people [27]]. Additionally, this population profile may suggest more concerns related to health issues in this group. Women are often more involved and engaged in community and family activities, which may contribute to increasing their sensitivity and perception of environmental health risks in their context [15]]. Furthermore, our sample was broadly similar to previous studies with regards to leptospirosis prevalence. An important outlier was the prevalence of our sample in RS, which was 19.8%, compared to 12% estimated in a previous study [1]]. This may have resulted in a higher awareness of the risks associated to leptospirosis and could have biased the results. The issues we encountered with the missing data could have affected these calculations and biased our final results.

PLOS Global Public Health

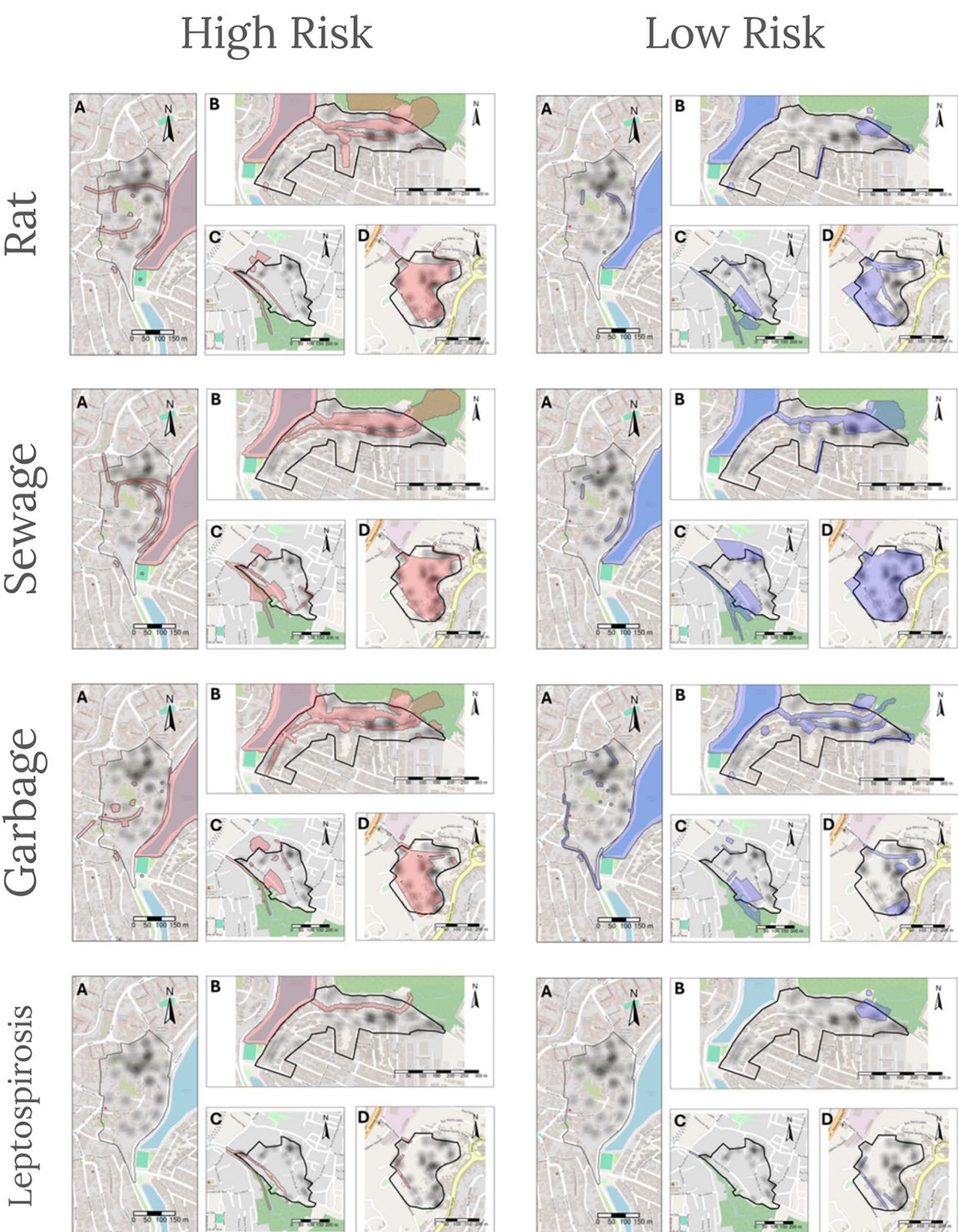

**Fig 2. Maps showing areas of high risk perception (red, left hand side) and low risk perception (blue, right hand side) for each of the high-lighted risks in each of the study areas: A) Alto do Cabrito, B) Marechal Rondon, C) Nova Constituinte and D) Rio Sena.** In an effort to main-tain anonymity, leptospirosis positive households are shown as a kernel density distribution denoted as a black-to-transparent graduating scale. The highest density is indicated by the darkest areas, which represent 4 households. The map used in the publication is not copyrighted, as it was created by the authors themselves using the free and open-source software **R**. The data used in preparing the map containing the boundaries of neighbor-ing countries, Brazil, Bahia, Salvador, and neighborhoods, are also open data and were downloaded from the website of the Instituto Brasileiro de

Geografia e Estatística (IBGE, 2021) and the Prefeitura Municipal de Salvador (Salvador, 2017). Both data can be consulted with their respective metadata on the websites: https://www.ibge.gov.br/geociencias/organizacao-do-territorio/malhas-territoriais/15774-malhas.html?=&t=downloads; http://mapeamento.salvador.ba.gov.br/geo/desktop/index.html#on=layer/default;bairros/bairros;scalebar_meters/scalebar_m;orto2016/Ortoimagem_Salvador_2016_2017&loc=76.43702828517625;-4278080;-1445884. The base layer of the maps is taken from OpenStreetMap which is made available at openstreetmap.org under the Open Database License (ODbL).

**Table 5. Number of positive rat traps and track plates within each type of rat risk perception. Corresponding Kappa score.**

| Rat Presence | Leptospirosis status | Risk Perception | |
|---|---|---|---|
| | | Rat | |
| | | High | Low |
| | Positive | 184 | 123 |
| | Negative | 238 | 71 |
| Kappa score | -0.17 | | |
| | p<0.001 | | |

**Table 6. Rubbish pile densities within each level of rubbish risk perception polygons.**

| Communities | Risk Perception | | | | Density difference (n/km2) |
|---|---|---|---|---|---|
| | Rubbish | | | | |
| | Low | | High | | |
| | N° | Density (n/km²) | N° | Density (n/km²) | |
| AC | 4 | 78.8 | 7 | 136.5 | 56.8 |
| MR | 2 | 27.2 | 24 | 260.4 | 233.2 |
| NC | 0 | 0 | 7 | 329.5 | 329.5 |
| RS | 1 | 88.2 | 7 | 162.8 | 74.6 |

Although we did not identify previous studies addressing the contribution of race/ethnicity to the perception of environmental health risk, it is important to consider the racial and socioeconomic inequalities experienced strongly by this population segment. Communities predominantly composed of Black populations, as in our study, are exposed to more environmental health risks, as they are historically subject to areas with inadequate housing conditions and more limited access to basic services [1,6]]. This aspect can influence both their perception of risk and their actual exposure to diseases like leptospirosis. Additionally, these populations generally have fewer material resources, and the conditions they live in increase their vulnerability to diseases.

The environmental characteristics of vulnerable communities also play a crucial role in exposure to risks that can cause leptospirosis. In this study, we found that residences located in the AC, MR, and RS communities were surrounded by more vegetation and soil but less impermeable terrain compared to the community average. This difference could offer a more favorable environment for the presence of rodents and could lead to increased contact with soil contaminated with urine from these hosts [[5,7]]. In contrast, the NC community presented a higher amount of impermeable terrain, which can significantly reduce exposure to contaminated soil and contact with *Leptospira*, especially during rainy periods. This could explain why perceptions in the NC community tend to be of low risks for the different factors. We also found that households in our study were closer to sewage and garbage points than the community average. Residences near sewage and garbage are generally at greater risk of exposure to pathogens, including the *Leptospira* bacterium [[5], [7]]. Residents' perceptions of these risks are, therefore, consistent with their reality, as many identified these factors as the main environmental health risks in their context.

Studies conducted in peripheral communities of Salvador identified the domestic rat as the main reservoir for *Leptospira* transmission [7]]. Additionally, rats sighted by residents were correlated with the risk of acquiring antibodies against *Leptospira* in a previous study [7]]. We found that residents understand the risks that rats pose, as they perceive them as a risk across all study areas. However, we found that there were more positive rat traps and track plates in areas where residents thought the risk of rat presence was low. Resident do correctly identify where rubbish piles accumulate—a more obvious risk, given the visual impact.

We also found a low perception of leptospirosis as a health risk. This is highlighted in the study area AC, where no participant identified leptospirosis as a health risk of any level. This suggests the need for health education actions addressing the modes of disease transmission. This is corroborated by a knowledge, attitudes and practices study performed in the MR community, where it was identified that residents presented some gaps regarding the mode of leptospirosis transmission [28]]. Our findings indicate that some factors associated to leptospirosis transmission are of greater concern to community residents than other factors and the disease itself. This could be due to the nature of these factors. In other words, residents are more aware of where rubbish accumulates given the visual impact, but are not as clear where rats are present. Here, we hypothesize that this may be related to the multiplicity of risks to which the population is exposed, which can contribute to some problems becoming "naturalized," leading to an underestimation of their presence in specific contexts, especially those further from homes or less visible. In the case of rats, the perception of low risk may be related to the fact that they do not settle in specific locations; they move according to the availability of food and avoid areas where they perceive danger, which makes their stable identification difficult and may contribute to the underestimation of that risk.

Future collaborative mapping studies should be concerned with the accuracy of data collection and explore in more detail the relationship between specific sociodemographic and environmental characteristics and their influence on the perception of environmental health risk. Additionally, there may be differences in the perception of environmental health risk according to the place of residence and proximity to risk in the communities where our study was conducted, which can be explored in other works. Furthermore, new studies should also focus on more effective strategies for interacting with public agencies and Geographic Information Systems (GIS) technologies that are more accessible and user-friendly to the community. This could increase the use, engagement, and sustainability of collaborative mapping processes facilitated by public agencies, reducing the consultation need for mediation by university groups. Gamification initiatives, such as the +Lugar platform [[29,30]] are being developed in this regard. In addition, research that evaluates the impact of interventions resulting from these collaborative processes on the communities involved is essential. Such research could investigate aspects such as the population of synanthropes in the environment, human health, and social control.

Although violence is not addressed in detail in this article, it represented a significant challenge for the study. Access to certain areas of the communities was restricted at times, which not only prevented mapping activities on those days but also hindered the participation of young partners in some planning meetings, as their movement on those days was curtailed. These restrictions did not impact the comprehensiveness of the collected data but affected the duration of some stages of the collaborative mapping process.

It is also important to highlight the positive effect of recruiting local young partners to be involved with the research. The participation of young partners not only increased the quality and relevance of the collected data but also promoted greater engagement and acceptance of the research by local communities. The direct involvement of young partners in the co-design process of field instruments and field research was fundamental to the success of the collaborative mapping. We recommend applying similar engagement practices in future community research projects.

Our study has limitations. The first is the cross-sectional design adopted, which does not allow for inferences of cause and effect relationships or testing of causal hypotheses. In addition, the presence of missing data for some study variables may have underestimated our findings. We also recognise that the work carried out using perceptions could be complemented by other qualitative techniques, such as focus groups or semi-structured interviews. An integrated

qualitative-quatitative approach, using these techniques alongside spatial methods of analysis, could help us understand more deeply the issues not captured in this study. Despite these limitations, this was one of the few studies, if not the first, to simultaneously assess the perception of environmental health risks through collaborative mapping, in four peripheral communities of Salvador, Brazil, with the support of young partners from these communities.

## Conclusion

In the current century, efforts to identify more effective and sustainable interventions for urban leptospirosis in vulnerable areas have been hampered by the lack of community involvement and participation in research and the production of information on environmental health risks. It is essential to prioritize the inclusion of communities in these contexts to ensure the relevance and applicability of proposed interventions, as well as their alignment with local priorities. We present a robust method to capture individual's perceptions of environmental health risks, focused on participatory research and collaborating with local youth partners. This presents an important method to involve communities in interventions affecting their health and wellbeing.

## Supporting information

**S1 Appendix. Maps comparing rat and rubbish risk perception with presence of each.**
(DOCX)

**S2 Appendix. Details on collaborative mapping data processing.**
(DOCX)

## Acknowledgments

We sincerely thank all the residents and community leaders of Marechal Rondon, Alto do Cabrito, Nova Constituinte, and Rio Sena. We also want to thank you for participating and trusting our research team. With everyone's collaboration and active involvement, this study was possible. To our research colleagues and field staff, our sincere thanks throughout all stages of the study. The work and collaboration of each of you were essential for collecting data and analyzing the information in this manuscript.

## Author contributions

**Conceptualization:** Pablo Ruiz Cuenca, Fabiana Almerinda G. Palma, Patricia Lustosa Brito, Federico Costa, Ricardo Lustosa.

**Data curation:** Pablo Ruiz Cuenca, Fabiana Almerinda G. Palma, Patricia Lustosa Brito, Marbrisa Nascimiento Reis das Virgens, Murilo Guerreiro Arouca, Ricardo Lustosa.

**Formal analysis:** Pablo Ruiz Cuenca, Fabiana Almerinda G. Palma, Patricia Lustosa Brito, Marbrisa Nascimiento Reis das Virgens, Murilo Guerreiro Arouca, Daiana Santos de Oliveira, Ricardo Lustosa.

**Funding acquisition:** Hussein Khalil, Yeimi Alexandra Alzete Lopez, Michael Begon, Federico Costa, Ricardo Lustosa.

**Investigation:** Pablo Ruiz Cuenca, Fabiana Almerinda G. Palma, Alexandre Mota Santos, Diogo Cesar de C. Santiago, Ricardo Lustosa.

**Methodology:** Pablo Ruiz Cuenca, Fabiana Almerinda G. Palma, Patricia Lustosa Brito, Marbrisa Nascimiento Reis das Virgens, Murilo Guerreiro Arouca, Alexandre Mota Santos, Daiana Santos de Oliveira, Diogo Cesar de C. Santiago, Emanuele Giorgi, Federico Costa.

**Project administration:** Hussein Khalil, Patricia Lustosa Brito, Federico Costa, Ricardo Lustosa.

**Resources:** Hussein Khalil, Yeimi Alexandra Alzete Lopez, Michael Begon, Federico Costa, Ricardo Lustosa.

**Software:** Pablo Ruiz Cuenca, Fabiana Almerinda G. Palma, Patricia Lustosa Brito, Marbrisa Nascimiento Reis das Virgens, Murilo Guerreiro Arouca, Ricardo Lustosa.

**Supervision:** Patricia Lustosa Brito, Yeimi Alexandra Alzete Lopez, Michael Begon, Emanuele Giorgi, Federico Costa, Ricardo Lustosa.

**Validation:** Patricia Lustosa Brito, Yeimi Alexandra Alzete Lopez, Michael Begon, Federico Costa, Ricardo Lustosa.

**Visualization:** Pablo Ruiz Cuenca, Fabiana Almerinda G. Palma, Patricia Lustosa Brito, Marbrisa Nascimiento Reis das Virgens, Murilo Guerreiro Arouca, Ricardo Lustosa.

**Writing – original draft:** Pablo Ruiz Cuenca, Fabiana Almerinda G. Palma, Murilo Guerreiro Arouca, Ricardo Lustosa.

**Writing – review & editing:** Pablo Ruiz Cuenca, Fabiana Almerinda G. Palma, Hussein Khalil, Patricia Lustosa Brito, Marbrisa Nascimiento Reis das Virgens, Murilo Guerreiro Arouca, Daiana Santos de Oliveira, Yeimi Alexandra Alzete Lopez, Diogo Cesar de C. Santiago, Michael Begon, Emanuele Giorgi, Federico Costa, Ricardo Lustosa.

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
