## [Decision Letter · Decision Letter 0]

27 Oct 2025

PGPH-D-25-02631

Collaborative Mapping: Perceived Environmental Risk of Leptospirosis in Urban Communities in Salvador, Brazil

Dear Dr. Ruiz Cuenca,

Thank you for submitting your manuscript to PLOS Global Public Health. After careful consideration, we feel that it has merit but does not fully meet PLOS Global Public Health’s publication criteria as it currently stands. Therefore, we invite you to submit a revised version of the manuscript that addresses the points raised during the review process.

The manuscript has been evaluated by two reviewers, and their comments are available below.

The reviewers have raised a number of major concerns. They feel the manuscript request improvements to the reporting of methodological aspects of the study, for example, regarding the how participants were selected and how the serological data collection was completed. Could you please carefully revise the manuscript to address all comments raised?

Please also ensure you upload copy of the questionnaire as Supporting Information when your submit your revised manuscript.

We look forward to receiving your revised manuscript.

Kind regards,

Katrien G. Janin, PhD

Staff Editor

Journal Requirements:

Additional Editor Comments (if provided):

Reviewers' comments:

Reviewer's Responses to Questions

**Comments to the Author**

1. Does this manuscript meet PLOS Global Public Health’s publication criteria? Is the manuscript technically sound, and do the data support the conclusions? The manuscript must describe methodologically and ethically rigorous research with conclusions that are appropriately drawn based on the data presented.? Is the manuscript technically sound, and do the data support the conclusions? The manuscript must describe methodologically and ethically rigorous research with conclusions that are appropriately drawn based on the data presented.

Reviewer #1: Partly

Reviewer #2: Partly

2. Has the statistical analysis been performed appropriately and rigorously?

Reviewer #1: No

Reviewer #2: No

3. Have the authors made all data underlying the findings in their manuscript fully available (please refer to the Data Availability Statement at the start of the manuscript PDF file)?

The PLOS Data policy requires authors to make all data underlying the findings described in their manuscript fully available without restriction, with rare exception. The data should be provided as part of the manuscript or its supporting information, or deposited to a public repository. For example, in addition to summary statistics, the data points behind means, medians and variance measures should be available. If there are restrictions on publicly sharing data—e.g. participant privacy or use of data from a third party—those must be specified.requires authors to make all data underlying the findings described in their manuscript fully available without restriction, with rare exception. The data should be provided as part of the manuscript or its supporting information, or deposited to a public repository. For example, in addition to summary statistics, the data points behind means, medians and variance measures should be available. If there are restrictions on publicly sharing data—e.g. participant privacy or use of data from a third party—those must be specified.

Reviewer #1: No

Reviewer #2: Yes

4. Is the manuscript presented in an intelligible fashion and written in standard English?

Reviewer #1: Yes

Reviewer #2: Yes

Reviewer #1: Dear Authors,

The manuscript is well written. However, there is a lack of method description. The results can do more in this analysis. I have some comments below.

1. Abstract: Please provide the full name of four areas.

2. Serological data: It is unclear how to define positive cases. How many serovars were tested? For the positive participants, were they positive for at least one serovar or something else?

3. Phase 2 Field surveys: How do the authors define the 244 residents (or participants)?

4. To understand the survey results, please provide the questionnaire.

5. Lines: 159-164, It is unclear how to define the type and level of risk as shown in Table 3.

6. Line: 181, What do the authors mean by eligible residents?

7. For the environmental characteristics, since the authors conducted the survey for 244 participants, who already have the address. I wonder if the authors can include the positive leptospirosis cases to identify the environmental risk rather than describe the characteristics. These results can support the discussion (lines: 338-347)

Reviewer #2: This is an interesting piece of research. The manuscript is well written, and the authors do a good job of presenting the rationale, methods, results, and conclusions. However, there are some weaknesses. The methods need more clarity, some additional statistics is needed, and the discussion should be developed further. See specific comments below.

1. Line 107: Change Familiar Income to Family Income in Figure 1.

2. Line 117: There are over 250 serovars of Leptospira in the word. The use of only 2 serovars in this study is not addressed.

3. Line 140: Append the questionnaire.

4. Line 177: Add a section in methods that identifies all software and packages used and cite these.

5. Line 136: What is the sampling frame? How were the 244 households chosen?

6. Results: The entire results section rely heavily on descriptive percentages, tests on significance is needed to make conclusions.

7. Lines 276-287: Should be moved to the methods section I indicated in point 4.

8. Line 309-314: Apart from vegetation, impervious terrain has been found to be significant risk factor for leptospirosis in urban environments as they flood more readily. Why was perceptions on flooding not considered in this study? In addition, if leptospirosis is mainly a disease of males and most of the participants in this study were females, the results would reflect the risk perceptions of females which may be different to males based on their roles and activities in the community.

9. Line 314-317: I am not sure what this statement is supposed to convey.

10. Line 324-325: How was your sample similar to the prevalence study. Was is the number of males and females? Because this appears to be a confounder that needs to be discussed.

11. Line 325-327: Is this outlier significant?

12. Line 339-341: Was this difference significant?

13. Line 344-346: As mentioned in point 8, impermeable terrain are high-risk flood areas and floods are a risk factor for leptospirosis. Is the prevalence of leptospirosis known in these regions?

14. Line 357-358: Any hypothesis as to why more rat positive areas were perceived as low risk for rats?

15. Lines 361-362: AC was also the region with the highest proportion of females. See comment 8 regarding disease perception and sex.

16. Lines 386-399: It is great to see these 2 paragraphs. Important points that need to be considered in such studies in the future.

**Do you want your identity to be public for this peer review?** For information about this choice, including consent withdrawal, please see our Privacy Policy..

Reviewer #1: No

Reviewer #2: **Yes:** Shahista NisaShahista NisaShahista NisaShahista Nisa

---

## [Decision Letter · Decision Letter 1]

25 Feb 2026

Collaborative Mapping: Perceived Environmental Risk of Leptospirosis in Urban Communities in Salvador, Brazil

PGPH-D-25-02631R1

Dear Pablo Ruiz Cuenca

We are pleased to inform you that your manuscript 'Collaborative Mapping: Perceived Environmental Risk of Leptospirosis in Urban Communities in Salvador, Brazil' has been provisionally accepted for publication in PLOS Global Public Health.

Best regards,

Megan Coffee, MD, PhD

Academic Editor

Reviewer Comments (if any, and for reference):

Reviewer's Responses to Questions

**Comments to the Author**

Reviewer #1: All comments have been addressed

publication criteria? Is the manuscript technically sound, and do the data support the conclusions? The manuscript must describe methodologically and ethically rigorous research with conclusions that are appropriately drawn based on the data presented.? Is the manuscript technically sound, and do the data support the conclusions? The manuscript must describe methodologically and ethically rigorous research with conclusions that are appropriately drawn based on the data presented.

Reviewer #1: Yes

3. Has the statistical analysis been performed appropriately and rigorously?

Reviewer #1: N/A

4. Have the authors made all data underlying the findings in their manuscript fully available (please refer to the Data Availability Statement at the start of the manuscript PDF file)?

The PLOS Data policy requires authors to make all data underlying the findings described in their manuscript fully available without restriction, with rare exception. The data should be provided as part of the manuscript or its supporting information, or deposited to a public repository. For example, in addition to summary statistics, the data points behind means, medians and variance measures should be available. If there are restrictions on publicly sharing data—e.g. participant privacy or use of data from a third party—those must be specified.requires authors to make all data underlying the findings described in their manuscript fully available without restriction, with rare exception. The data should be provided as part of the manuscript or its supporting information, or deposited to a public repository. For example, in addition to summary statistics, the data points behind means, medians and variance measures should be available. If there are restrictions on publicly sharing data—e.g. participant privacy or use of data from a third party—those must be specified.

Reviewer #1: No

5. Is the manuscript presented in an intelligible fashion and written in standard English?

Reviewer #1: Yes

Reviewer #1: Thank you for the revision. I have no further comments.

**Do you want your identity to be public for this peer review?** For information about this choice, including consent withdrawal, please see our Privacy Policy..

Reviewer #1: No
